# User Perceptions of Avatar-Based Patient Monitoring for Intensive Care Units: An International Exploratory Sequential Mixed-Methods Study

**DOI:** 10.3390/diagnostics13213391

**Published:** 2023-11-06

**Authors:** Justyna Lunkiewicz, Greta Gasciauskaite, Tadzio Raoul Roche, Samira Akbas, Christoph B. Nöthiger, Michael T. Ganter, Patrick Meybohm, Sebastian Hottenrott, Kai Zacharowski, Florian Jürgen Raimann, Eva Rivas, Manuel López-Baamonde, Elisabeth Anna Beller, David Werner Tscholl, Lisa Bergauer

**Affiliations:** 1Department of Anesthesiology, University Hospital Zurich, Raemistrasse 100, 8091 Zurich, Switzerland; justyna.lunkiewicz@usz.ch (J.L.); tadzioraoul.roche@usz.ch (T.R.R.); samira.akbas@usz.ch (S.A.); christoph.noethiger@usz.ch (C.B.N.); elisabeth.beller0@icloud.com (E.A.B.);; 2Institute of Anesthesiology and Critical Care Medicine, Clinic Hirslanden Zurich, 8032 Zurich, Switzerland; michael.ganter@hirslanden.ch; 3Department of Anesthesiology, Intensive Care, Emergency and Pain Medicine, University Hospital Wuerzburg, University of Wuerzburg, 97070 Wuerzburg, Germany; meybohm_p@ukw.de (P.M.); hottenrott_s@ukw.de (S.H.); 4Department of Anesthesiology, Intensive Care Medicine and Pain Therapy, University Hospital Frankfurt, Goethe University Frankfurt, 60629 Frankfurt, Germany; 5Department of Anesthesiology, Intensive Care Medicine and Pain Therapy, Hospital Clinic of Barcelona, University of Barcelona, 08007 Barcelona, Spain; erivas@clinic.cat (E.R.); lopez10@clinic.cat (M.L.-B.)

**Keywords:** visual patient avatar, patient monitoring, situation awareness, human factors, user-centered design, user perception

## Abstract

Visual Patient Avatar ICU is an innovative approach to patient monitoring, enhancing the user’s situation awareness in intensive care settings. It dynamically displays the patient’s current vital signs using changes in color, shape, and animation. The technology can also indicate patient-inserted devices, such as arterial lines, central lines, and urinary catheters, along with their insertion locations. We conducted an international, multi-center study using a sequential qualitative-quantitative design to evaluate users’ perception of Visual Patient Avatar ICU among physicians and nurses. Twenty-five nurses and twenty-five physicians from the ICU participated in the structured interviews. Forty of them completed the online survey. Overall, ICU professionals expressed a positive outlook on Visual Patient Avatar ICU. They described Visual Patient Avatar ICU as a simple and intuitive tool that improved information retention and facilitated problem identification. However, a subset of participants expressed concerns about potential information overload and a sense of incompleteness due to missing exact numerical values. These findings provide valuable insights into user perceptions of Visual Patient Avatar ICU and encourage further technology development before clinical implementation.

## 1. Introduction

In an intensive care unit (ICU), situation awareness and prompt response to patient status changes are crucial, particularly when looking after multiple potentially deteriorating patients [1,2]. With higher patient loads and an aging population, the patient-to-care provider ratio, particularly for nursing staff, has significantly increased [3]. In such circumstances, the risk of not promptly perceiving a potential problem and taking appropriate action raises the chances of medical errors and, consequently, patient morbidity and mortality [1,3,4,5]. Situation awareness plays a crucial role in medical settings. It is described as the cognitive process of perceiving, comprehending, and projecting the clinical situation, enabling healthcare professionals to make informed decisions and take timely actions to ensure patient safety and optimal outcomes [2,6,7]. The concept of situation awareness provides a framework for developing user-centered design systems [8]. Instead of simply presenting raw information, user-centered design integrates data in a way that aligns with users’ abilities and needs. This approach is motivated by the goal of achieving optimal functioning of the human–machine interaction as a whole and ensuring safety [8,9].

An example of user-centered visualization technology is Philips Visual Patient Avatar [10,11,12]. It is an innovative approach to patient monitoring that is designed to enhance user situation awareness [10,11]. This technique transforms alphanumeric monitoring data into a visual format represented by the avatar. Philips Visual Patient Avatar dynamically displays the patient’s current vital signs using modifications in color, shape, and animations. This technology is guided using a user-centered design philosophy inspired by Endsley [9], logical principles from Wittgenstein’s Tractatus Logico-Philosophicus [13], and insights from human–computer interaction as described in NASA’s publication “On Organization of Information: Approach and Early Work” by Degani et al. [14]. Since March 2023, this technique has been integrated into Philips^®^ IntelliVue MX patient monitors [15]. This approach enables a more efficient perception of vital signs [10,16] and increases the likelihood of verbalizing the cause of the emergencies [17] compared to conventional monitoring. Furthermore, computer-based studies showed that Phillips Visual Patient Avatar significantly improves diagnostic confidence among anesthetists and reduces workload compared to standard monitoring [10,16,17,18,19].

An extended version of Visual Patient Avatar, called Visual Patient Avatar ICU, is currently being developed and investigated. Visual Patient Avatar ICU can additionally display patient-inserted devices such as arterial, central lines, and urinary catheters along with their respective insertion locations (Figure 1) [19]. A previous computer-based study has demonstrated that the Visual Patient Avatar ICU improves information transfer, enhances diagnostic confidence, and reduces the perceived workload of ICU staff compared to conventional monitor modalities [19]. 

This study aimed to evaluate user perception of Visual Patient Avatar ICU among physicians and nurses in multiple intensive care units across various international hospitals. The feedback gathered from critical care staff helps identify the strengths and potential for improvement in this newly developed technology and contributes to its further improvement before clinical implementation.

## 2. Materials and Methods

### 2.1. Approval and Consent

Based on the local regulations in Switzerland, Germany, and Spain, ethical approval was not required, as this study did not fall within the scope of the Human Research Act. The leading ethics committee in Zurich, Switzerland, provided a declaration of non-jurisdiction. However, all participants provided written consent for recording, statistical analysis, and data publication.

### 2.2. Study Design

This was an international, multi-center, researcher-initiated study utilizing an exploratory sequential mixed-methods design. The study was conducted across five tertiary care hospitals, namely the University Hospital of Zurich and Hirslanden Clinic of Zurich in Switzerland, the University Hospital of Frankfurt and University Hospital of Wuerzburg in Germany, and the Hospital Clinic de Barcelona in Spain. We interviewed the participants between June and August 2021. The online survey was conducted between July and October 2021. In this study, we included the same 50 participants who were part of a previous computer-based study that compared two different patient monitoring modalities (Visual Patient Avatar ICU versus conventional monitor) [19]. 

### 2.3. Previous Study

In the previous computer-based study, a total of 50 participants, consisting of five ICU nurses and five physicians at each of the five study sites, engaged with five distinct patient scenarios [19]. These scenarios were presented twice, once as Visual Patient Avatar ICU and once as the conventional modality, resulting in a total of ten cases per participant. This study aimed to examine the impact of Visual Patient Avatar ICU on information transfer, which was assessed by accuracy in evaluating vital signs and installations [19]. Subsequent to the completion of all cases, structured interviews were conducted as an integral component of this mixed-methods study.

### 2.4. Participant Interviews and Online Survey

#### 2.4.1. Part I: Participant Interviews

In the interview phase of our study, we requested input from all participants regarding two questions: “What did you like about Visual Patient Avatar ICU?” and “What did you dislike about Visual Patient Avatar ICU?” directly after they participated in the computer-based study. We aimed to investigate the positive and negative perceptions of Visual Patient Avatar ICU immediately after its use. Participants were instructed to record their thoughts as field notes using an iPad (Apple Inc., Cupertino, CA, USA), utilizing two separate text boxes. There was no specified time constraint, and we accommodated responses in German, Spanish, or English.

In addition, a brief demographic survey was conducted to collect participant characteristics. To commence the systematic analysis, the field notes provided by participants in German and Spanish were translated into English using the online translation service DeepL (DeepL GmbH, Cologne, Germany) (Appendix A). The translated results underwent individual assessment to ensure plausibility and accuracy. Subsequently, the two authors, EAB and LB, organized the translated field notes into separate statements based on content topics. The authors (EAB, SA, LB) identified recurring patterns in responses and developed a coding tree that encompassed the major topics and subthemes addressed by the participants (Figure 2) [20]. EAB and LB independently assigned all statements to the coding tree. To validate this process, percentage agreement and inter-rater reliability calculations were conducted. In divergent allocations, the two raters engaged in discussions and reached a mutual agreement to determine the appropriate coding.

#### 2.4.2. Part II: Online Survey

For the second phase of the study, we formulated five statements based on the major topics identified using qualitative assessment. The statements were: Visual Patient’s ICU installation display is helpful. Visual Patient ICU provides a better overview. Sometimes, the Visual Patient ICU is overloaded with information. Although Visual Patient ICU is intuitive, I would need training before clinical use. In a critical situation, Visual Patient ICU enabled me to recognize pathological deviations more quickly. An online questionnaire was created using SurveyMonkey (SVMK Inc., San Mateo, CA, USA), and the corresponding link was emailed to all previous participants, inviting them to evaluate the statements on a 5-point Likert Scale ranging from strongly disagree to strongly agree. Additionally, we collected anonymous demographic data. Participation in the online survey required less than five minutes. A single reminder to complete the questionnaire was sent after ten days. After an additional week, the survey portal was closed, concluding the data collection process. To ensure consistent timing, we contacted the different study centers staggered, maintaining approximately 40-day intervals between the qualitative and quantitative phases of the study.

### 2.5. Statistical Analysis

Data management and creation of figures were performed with Microsoft Word, Excel, and PowerPoint (Microsoft Corporation, Redmond, Washington, DC, USA). In the qualitative study section, we present the number of statements and their respective percentages in relation to the total number of statements and major topics. To address the consistency of EAB and LB ratings, we calculated the percent agreement and inter-rater reliability using Cohen’s kappa in R version 4.0.5 (R Foundation for Statistical Computing, Vienna, Austria) [21,22]. The results of the quantitative study section are reported as medians with interquartile ranges. We used the one-sample Wilcoxon signed-rank test (IBM SPSS Statistics 26, International Business Machines Corporation, Armonk, New York, NY, USA) to evaluate the symmetry of the answer distribution to the given statements around the median (representing the neutral response) and to determine any tendency towards agreement or disagreement. Statistical significance was defined as *p* < 0.05.

## 3. Results

### 3.1. Study and Participant Characteristics

We recruited 25 nurses and 25 physicians from ICUs to participate in structured interviews. Of these, 40 participants (80%) later responded to the email invitation and completed the online survey. The characteristics of the participants are provided in Table 1.

### 3.2. Part I: Participant Interviews

#### 3.2.1. Coding Template

By analyzing the field notes obtained, we derived a total of 148 statements. Using an inductive free coding approach, we identified four major positive themes and three major negative themes [20] (Figure 2). Using the coding template, the percentage agreement between EAB and LB in assigning the 148 statements was 82.4%. The calculated inter-rater reliability, indicated by Cohen’s kappa of 0.802, demonstrated substantial agreement [23,24].

Despite several discussions, 9 out of the 148 statements (6%) remained unclear in terms of their meaning or intention and were classified as “not codable”. After this process was completed, the percentage agreement reached 100%, and the remaining 139 statements were used as a reference for further calculations.

Overall, participants expressed moderately more positive statements (78/139, 56%) than statements regarding their dislikes of the Visual Patient Avatar ICU (61/139, 44%). Figure 3 provides an overview of the distribution of all statements derived from the field notes commenting on Visual Patient Avatar ICU. Table 2 outlines the major themes with their hierarchical subthemes and the corresponding percentages and examples. In the subsequent section, we provide a detailed description of the categories and participants’ perceptions.

#### 3.2.2. Positive Statements about Visual Patient Avatar ICU

##### Design

Of the 139 statements analyzed, 29 (21%) were related to positive design features of Visual Patient Avatar ICU. We further categorized this major topic into two subthemes: “Overview”, comprising 14 of the 139 statements (10%), and “Illustration”, comprising 15 of the 139 statements (11%). Participants appreciated the presentation of information in a simple overview format, as it allowed them to grasp everything at a glance (participant #8). Regarding the Visual Patient Avatar ICU illustration, the use of colored markings (participant #6) was frequently mentioned as a positive design feature. Additionally, participants, such as Participant #12, expressed their liking for the visualization of organ systems.

##### Intuitiveness

Within the analyzed statements, 17 out of 139 (12%) highlighted characteristics such as simple handling and easy understanding (participant #26) associated with Visual Patient Avatar ICU. These findings were consolidated under the major topic “Intuitive” with agreement between the raters. Participant #7 mentioned that the system facilitated better retention of information, while Participant #3 emphasized a quick learning curve in interpreting the Visual Patient Avatar ICU.

##### Time Saving 

We identified “Time saving” as another major topic, comprising 16 out of 139 (12%) statements. Participant #9 stated that serious problems were clearly presented immediately, enabling quick problem identification (participant #50).

##### Patient Inserted Devices

Among the positive statements, the feature of displaying patients’ installations was frequently highlighted, leading us to define it as another major topic, with 16 out of 139 (12%) statements. Participant #10 mentioned that the installations were immediately clear, making it easier for some participants to memorize the catheters (participant #5).

#### 3.2.3. Negative Statements about Visual Patient Avatar ICU

##### Design

The design of the Visual Patient Avatar ICU received critical feedback in 40 out of 139 statements (29%), which we identified as a major topic. Two distinct subthemes emerged, namely “Overload” and “Illustration”, accounting for 24 out of 139 statements (17%) and 16 out of 139 statements (12%), respectively. Several participants raised concerns about overlapping information (participant #18) and a crowded visual representation (participant #21), resulting in sensory overload during initial exposure (participant #27). Moreover, specific aspects of the illustration were brought up in various statements. For instance, Participant #17 expressed the opinion that the graphics could benefit from a more professional appearance. Participant #12 commented on the lack of visual impact in the representation of the tidal volume, while Participant #16 criticized the central venous pressure visualization.

##### Unfamiliarity

Concerning the negative aspects of the Visual Patient Avatar ICU, the participants’ responses indicated a prevalent perception of “Unfamiliarity”, with 16 out of 139 statements (12%) addressing this issue. It became evident that similar to Participant #35, who mentioned that “It takes time to get used to it”, other participants also asserted the need for “A period of habituation” (participant #39) or emphasized the necessity for “more practice” (participant #49).

##### Incompleteness

Among the 139 statements analyzed, 5 (4%) expressed concerns regarding “Incompleteness”, which was identified as an additional major topic using inductive free coding. Participant #4 mentioned missing numbers as a specific example of this incompleteness. In another statement, Participant #20 critically questioned the level of saturation, asking, Saturation low: how low is it? 90% or 70%?”

### 3.3. Part II: Online Survey

The results of the assessment of the five statements, based on the major topics identified during the qualitative analysis of the field notes, are graphically represented in Figure 4. 

## 4. Discussion 

### 4.1. Principal Findings

Visual Patient Avatar ICU received positive feedback regarding its design features, intuitiveness, time-saving aspect, and the display of patient-inserted devices. Participants appreciated the simple overview format, color markings, and visualization of organ systems. The system was perceived as easy to understand and facilitated information retention.

On the other hand, concerns were raised about crowded visual presentations and sensory overload. Unfamiliarity was a prevalent perception, emphasizing the need for time and practice to become accustomed to the system. Incompleteness was also mentioned, particularly regarding missing information and clarity of certain parameters. 

Our data are consistent with previous studies investigating user perception of Visual Patient Avatar among anesthesiology personnel. One of the first studies researching Visual Patient Avatar monitoring in 2018 highlighted that over 80% of anesthesia providers found it intuitive and easy to learn [11,25]. Furthermore, study participants could correctly identify more than 70% of all vital signs visualizations without any prior training [15], confirming our results that users find Visual Patient Avatar intuitive. This further means that a Visual Patient Avatar can be easily thought of in a classroom-based setting. Rössler et al. compared different teaching settings for Visual Patient Avatar instructions on a one-to-one basis or classroom-based instructions [26]. Although one-to-one instructions were most effective, the class instructions showed also to be a highly effective teaching model. However, one needs to recognize the need for some critical care professionals to work over an extended period with the new visualization technology to have the feeling of familiarity and routine in daily practice. 

The extended version of Visual Patient Avatar, Visual Patient Avatar ICU, additionally displays patient-inserted devices such as arterial and central lines, providing critical care staff with a better overview of a patient. This can provide critical care staff with a better overview of a patient. The authors believe that it may also potentially facilitate handover and prevent the omission of certain patient-inserted devices. The routine check of patient-inserted devices for signs of infection, change of dressing, and recognizing the extended period of the device in situ requiring removal is highly crucial and prevents nosocomial infections and subsequent complications [27,28,29,30,31]. The first step in care for patient-inserted devices care is the awareness that these are present. 

On the other hand, extra visualizations providing us with valuable additional information may contribute to information overload. These have been described as “sensory overload during initial exposure” or “crowded” by less than 20% of study participants. There is a fine balance to find in providing medical staff with compact information and, at the same time, not overwhelming the clinical picture. Interestingly, Visual Patient Avatar ICU in its current form has been found to lower perceived workload in computer-based settings [19]. This implies that although to some participants, it may appear crowded, it is still simplifying the process of comprehension of patient monitoring. Furthermore, Görges et al. demonstrated that enhanced patient monitoring with additional information resulted in shorter median decision-making times, improved nurse triaging, and reduced frustration by implementing a broader perspective on ICU monitoring [1].

Another issue worth addressing is a sense of incompleteness that has been expressed in a qualitative part of our study. Missing numbers in Visual Patient Avatar ICU were for a few study participants, leading to uncertainty. This is consistent with data from other studies that investigated three different monitoring systems: Visual Patient Avatar, conventional monitor, and split screen, combining both modalities [32]. The positive response to the split screen monitoring system was that it provides a symbiotic modality that helps to focus on vital parameter changes, first spotting it on Phillips Visual Patient Avatar, followed by quantification on conventional monitoring and, thus, increasing its safety [32].

### 4.2. Strengths and Limitations

This study has its strengths and limitations. It was a multi-center, international study across three European countries, including large university hospitals. This design should reduce selection bias and confounding factors for intercultural differences, diverse clinical settings, and healthcare systems.

Qualitative research involves analyzing non-numeric data to uncover overall trends and deeper meanings in individual responses without assigning frequencies to the issues identified in the data [33]. This leads to equal attention to rare and frequently described points. Furthermore, the inability to test for statistical significance reduces the confidence in data generalizability compared to quantitative data [33]. To compensate for this, we performed a sequential mixed-methods study: the conclusions from the initial qualitative study served as the foundation for the second quantitative phase, which assessed the consistency and generalizability of the findings. Combining qualitative and quantitative approaches in a mixed-methods design allowed for the examination of complex phenomena and produced more robust results than using either method alone.

Furthermore, the study participants’ recruitment was not purely random and was dependent on the doctors’ and nurses’ availability and schedule of the initial simulation study [21]. There is, however, little reason to believe that staff availability would introduce significant bias for user perception of Visual Patient Avatar ICU.

This is the first time the Visual Patient Avatar ICU concept was examined and provided the general idea for the initial reactions of medical professionals towards it. Due to the small sample size and 80% response rate to the online survey, the results need to be interpreted with caution. The results provide a general idea of user perception and should be interpreted more as a result of the pilot study. Further studies are needed with a larger sample size to fully investigate user perception of Visual Patient Avatar ICU.

## 5. Conclusions

This was the first computer-based study to evaluate the user perception of Visual Patient Avatar ICU among physicians and nurses in multiple intensive care units across various international hospitals. The feedback gathered from critical care staff helps to identify the strengths and potential for improvement in this newly developed technology and contributes to its further development.

Overall, critical care professionals expressed a positive attitude towards Visual Patient Avatar ICU, describing it as an easy and intuitive tool that enhances information retention and facilitates problem identification. Participants appreciated the presentation of information in a simple overview format, as it allowed them to grasp everything at a glance. However, a subset of participants raised concerns about potential information overload, initial unfamiliarity with the technology, and a sense of incompleteness when patient monitoring numbers were not included. These findings offer valuable insights into the user perception of Visual Patient Avatar ICU and encourage further development before its clinical implementation.

## Figures and Tables

**Figure 1 diagnostics-13-03391-f001:**
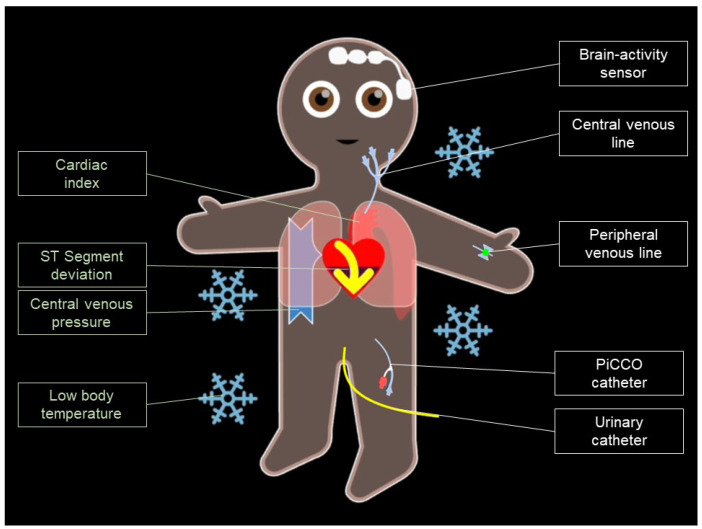
Exemplary presentation of Visual Patient ICU. The avatar’s eyes are open, indicating a high bispectral index as measured using the brain-activity sensor installed on the forehead. A central venous catheter is inserted into the left jugular vein, and the animation of the vena cava symbolizes the central venous pressure. Additionally, the VP ICU avatar has a PiCCO catheter inserted to measure the cardiac index. This is illustrated by the schematic representation of the aorta and the amount of red blood cells being ejected. The avatar’s beige skin tone represents a normal peripheral oxygen saturation. The yellow arrow within the heart corresponds to the connected electrocardiogram. The lower half of the heart being darkened indicates the presence of a detected ST-Segment deviation. Finally, the presence of ice crystals surrounding the avatar symbolizes a low patient temperature.

**Figure 2 diagnostics-13-03391-f002:**
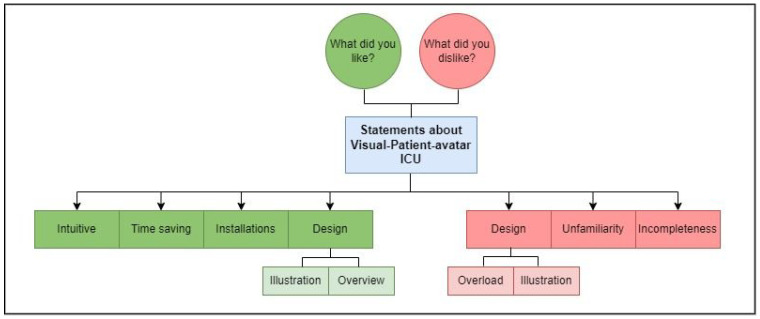
The coding tree depicts the positive and negative perceptions of the Visual Patient ICU. Using word counts and inductive coding, we identified the main themes along with their corresponding subthemes. The data for this analysis came from field notes collected from a total of 50 participants.

**Figure 3 diagnostics-13-03391-f003:**
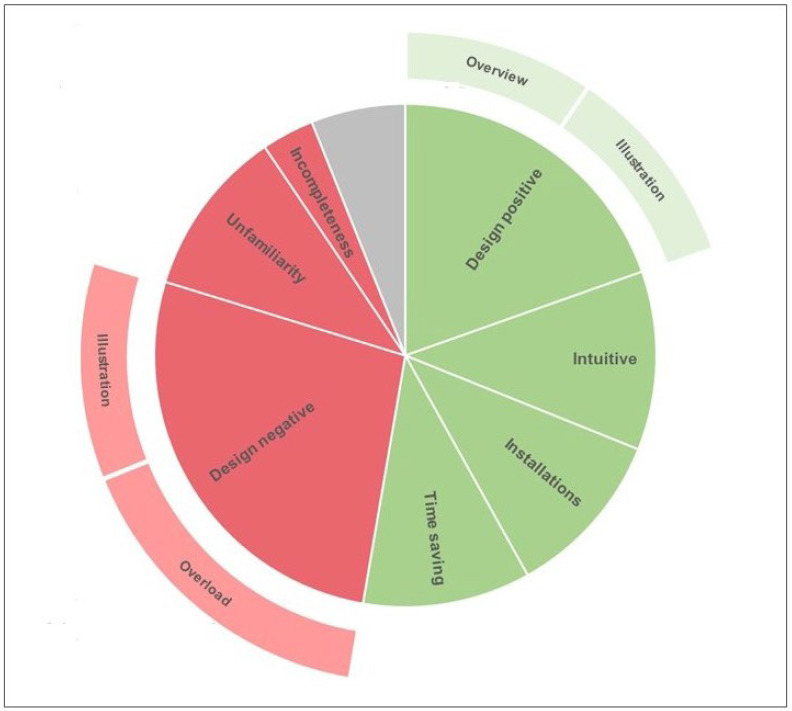
This two-level chart depicts the distribution of statements derived from field notes. The innermost circle encompasses the seven major topics, with a designated “not codable” section represented by a grey field. Moving towards the outer circle, the associated subthemes are displayed. The width of each section indicates the corresponding percentage of statements relating to the topic out of the total statements collected (*N* = 148).

**Figure 4 diagnostics-13-03391-f004:**
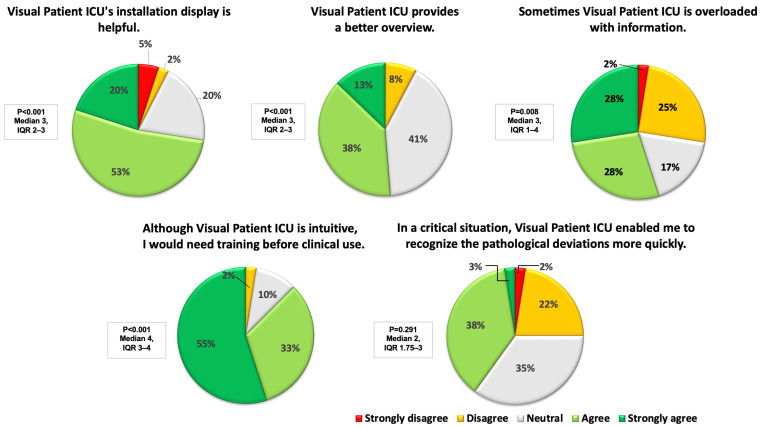
Graphical presentation of the online survey ratings as half doughnut charts. Results are shown as median and interquartile range (IQR). One-sample Wilcoxon signed-rank tests were used to determine whether the answers differed from neutral (*p* < 0.05). *N* = 40 for each statement.

**Table 1 diagnostics-13-03391-t001:** Study and participant characteristics.

	Part I: Participant Interviews	Part II: Online Survey
Period of data collection	23 June 2021–27 August 2021	28 July 2021–15 October 2021
Total number of participants	50	40
Number of nurses (%)	25 (50%)	21 (52.5%)
Number of physicians (%)	25 (50%)	19 (47.5%)
Number of female participants (%)	19 (38%)	17(42.5%)
Median (IQR) age in years	37.0 (33.0–43.8)	37 (32.75–42.75)
Median (IQR) work experience in years	10.5 (7.2–16.8)	10 (7–17.25)

**Table 2 diagnostics-13-03391-t002:** The major topics and subthemes with counts, percentages, and selected examples. The field notes obtained from 50 participants gave a total of 148 statements. The percentages refer to all codable statements (*N* = 139).

Major Topic	Subtheme	Examples
Design positive 29 of 139 (21%) statements	Overview 14 of 139 (10%) statements	Participant #15: Better overview. Participant #30: You capture a lot of information at a glance.
	Illustration positive 15 of 139 (11%) statements	Participant #12: Possibility to recognize the ST elevation by color. Participant #40: Situations are color-coded.
Intuitive 17 of 139 (12%) statements		Participant #26: Easy understanding. Participant #42: It is much more intuitive.
Time saving 16 of 139 (12%) statements		Participant #1: Deviations can be detected more quickly. Participant #8: You can quickly see if everything is okay or not okay.
Installations 16 of 139 (12%) statements		Participant #10: Installations immediately clear. Participant #43: Cool for lines and devices.
Design negative 40 of 139 (29%) statements	Overload 24 of 139 (17%) statements	Participant #5: Too much information in one image. Participant #38: Sometimes too confusing.
	Illustration negative 16 of 139 (12%) statements	Participant #17: The graphics could look more professional. Participant #41: Some parameters, like arterial blood pressure are difficult to see.
Unfamiliarity 16 of 139 (12%) statements		Participant #35: It takes time to get used to it. Participant #46: Training needed.
Incompleteness 5 of 139 (4%) statements		Participant #20: Saturation low: how low is it? 90% or 70%? Participant #44: Omits all the information that gives us the morphology of the curves.
Not codable 9 of 148 (6%) statements		Participant #1: Nothing. Participant # 23: Great, if it works.

## Data Availability

The datasets used and/or analyzed during the current study are available from the corresponding author upon reasonable request.

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
