# Peer review of "User Perceptions of Avatar-Based Patient Monitoring for Intensive Care Units: An International Exploratory Sequential Mixed-Methods Study"

_diagnostics, 2023, doi:10.3390/diagnostics13213391_

Round 1
Reviewer 1 Report
Comments and Suggestions for Authors
This work is well within the scope of the Diagnostics and it may be of interest to most of the readers of this journal.
It was clear that this study aimed to evaluate user perception and acceptance of Visual Patient Avatar 86 ICU among physicians and nurses in multiple intensive care units across various international hospitals.
The manuscript shows introductory background material sufficient for someone not an expert in this area to understand the context and significance of this work, with good references to follow, especially in the field of Visual Patient Avatar 86 ICU in vivo.
It proposes multimodal and multi-scale approaches to user-centered visualization technology.
For this study, the researchers recruited 25 nurses and 25 physicians to participate in structured interviews. Only 40 participants responded to the email invitation and completed the online survey. Why was the participation rate so low? Wasn't a preparation made in advance to ensure the participation to the end of as many participants as possible?
It is understood that the number of 50 participants is small anyway. When there are such high abstinence rates, any attempt to explain the result is risky.
Apart from the lack of explanation for the large abstention rate, the conclusions are even more ambiguous because as recorded in the properly designed statistical analysis, it appears that 56% of the participants expressed moderate and more positive statements, while 44% declared their dislike in relation to Visual Patient Avatar. So the results of those who answered do not show a significant positive contribution to the result of the use of this technology.
My personal opinion is that the contribution is very large, but this does not result from such a small sample and from a very small training of the staff, which obviously in combination with the lack of staff numerically was probably done with a lot of effort by the participants. Ιn the actual conditions lack of staff and work intensification, we all know how difficult they are.
Anyway, the conclusions describe a different goal than the original one. That is, it states that "the main objective of the research was to identify the strengths and improvement possibilities of this innovative monitoring technology and to contribute to its further improvement before its application in clinical settings". At the same time, the introduction was described that: "This study aimed to assess the user perception and acceptance of the Visual Patient Avatar ICU among physicians and nurses in multiple intensive care units in several international hospitals."
Obviously, the contribution of new technologies is particularly important and can work very positively and in various ways in improving the health provided and upgrading the quality of life, but the results of this research do not clearly prove this.
So, even if you mention the limitations and the aim of this research in the relevant sections, it would be good to see more clearly that because of the small number of total participants, the conclusions could work in principle as guidelines until they are verified by other researchers with a larger number of participant. Also, all the required advantages of properly evaluating a new technology should be ensured. That is, a sufficient number of health personnel, appropriate and comprehensive training and initially providing more time to the study of all those parameters presented comprehensively with the new technology. Finally, it would be very interesting if we had a statistical result regarding the survival statistics of patients from hospitals that use these technologies and from others that operate more traditionally.
I believe that although the advantages of this technology were not clearly seen, I would strongly agree with the authors of the article that the findings offer valuable insights into user perception of the Visual Patient Avatar ICU and other similar tools, thus encouraging further development before its clinical application, for this reason, I will accept the research with the above observations.
In conclusion, this article and its scientific conclusions are interesting.
Turnitin returned a similarity index of 26% without a bibliography (33% with) and quotes, so it’s a little bit high, according to Turnitin it shows marginally high plagiarism. At the same time, the AI writing report contains the total percentage of prose sentences contained in a long form of writing within the submitted document that Turnitin's model identifies as AI-generated as 0%.
On the whole, the English in this manuscript is very good.
For all the above, I have opted to recommend a Minor Revision.
Specific comments
Materials and Methods
P3, L92: ‘The Materials and Methods should be described with sufficient details to allow’ à Please rephrase.
Comments on the Quality of English LanguageOn the whole, the English in this manuscript is very good.
Author Response
Dear Reviewer,
Thank you for the valuable and constructive feedback. Hereby I respond to the reviewers’ comments and resubmit the revised manuscript having incorporated required adjustments.
Reviewer 1:
Feedback nr 1:
It proposes multimodal and multi-scale approaches to user-centered visualization technology.
For this study, the researchers recruited 25 nurses and 25 physicians to participate in structured interviews. Only 40 participants responded to the email invitation and completed the online survey. Why was the participation rate so low? Wasn't a preparation made in advance to ensure the participation to the end of as many participants as possible?
Author`s reply: Thank you for recognizing this weakness in our work. We also agree that it would be significantly better if all study participants replied to an online survey. Due to feasibility, consistency of the methodology across all centers and the international nature of the study, we maintained 40-day interval between the qualitative and quantitative phases of the study. All participants were clearly informed in advance about the study protocol and about the online survey in due course. As study participants were clinicians and, despite the initial commitment to study participation, reply to online survey over time may have been more difficult due to clinical duties. To improve our response rate a reminder to complete the questionnaire was sent after ten days. The response rate of 80% in comparison to similar studies described in the literature, although could be better, could still be considered as a valid and representative response rate.
VanGeest J, Johnson TP. Surveying Nurses: Identifying Strategies to Improve Participation. Evaluation & the Health Professions. 2011;34(4):487-511. doi:10.1177/0163278711399572
Feedback nr 2:
It is understood that the number of 50 participants is small anyway. When there are such high abstinence rates, any attempt to explain the result is risky.
Apart from the lack of explanation for the large abstention rate, the conclusions are even more ambiguous because as recorded in the properly designed statistical analysis, it appears that 56% of the participants expressed moderate and more positive statements, while 44% declared their dislike in relation to Visual Patient Avatar. So the results of those who answered do not show a significant positive contribution to the result of the use of this technology.
My personal opinion is that the contribution is very large, but this does not result from such a small sample and from a very small training of the staff, which obviously in combination with the lack of staff numerically was probably done with a lot of effort by the participants. Ιn the actual conditions lack of staff and work intensification, we all know how difficult they are.
Anyway, the conclusions describe a different goal than the original one. That is, it states that "the main objective of the research was to identify the strengths and improvement possibilities of this innovative monitoring technology and to contribute to its further improvement before its application in clinical settings". At the same time, the introduction was described that: "This study aimed to assess the user perception and acceptance of the Visual Patient Avatar ICU among physicians and nurses in multiple intensive care units in several international hospitals."
Obviously, the contribution of new technologies is particularly important and can work very positively and in various ways in improving the health provided and upgrading the quality of life, but the results of this research do not clearly prove this.
So, even if you mention the limitations and the aim of this research in the relevant sections, it would be good to see more clearly that because of the small number of total participants, the conclusions could work in principle as guidelines until they are verified by other researchers with a larger number of participant.
Also, all the required advantages of properly evaluating a new technology should be ensured. That is, a sufficient number of health personnel, appropriate and comprehensive training and initially providing more time to the study of all those parameters presented comprehensively with the new technology.
Author`s reply: Thank you for your valuable observation. This is a first time when Visual Patient ICU concept was examined and provided general idea for initial reactions of medical professional towards it. We are in agreement that due to small sample size, results need to be interpreted with caution. The results, both positive and negative comments, bring us one step closer to understanding how medical professionals perceive and benefit from VP ICU. However, further studies are needed with larger sample size to fully investigate user perception of Visual Patient ICU. We have now also addressed the issue with small sample size and response rate to the online survey of 80% in limitation section in our discussion (lines 455 – 460). Furthermore, the goal of the study was rephrased and checked for consistency across all sections of the manuscript.
Feedback nr 3:
Finally, it would be very interesting if we had a statistical result regarding the survival statistics of patients from hospitals that use these technologies and from others that operate more traditionally.
I believe that although the advantages of this technology were not clearly seen, I would strongly agree with the authors of the article that the findings offer valuable insights into user perception of the Visual Patient Avatar ICU and other similar tools, thus encouraging further development before its clinical application, for this reason, I will accept the research with the above observations.
Specific comments
Materials and Methods
P3, L92: ‘The Materials and Methods should be described with sufficient details to allow’ à Please rephrase.
Author`s reply:
Thank you for your constructive feedback and valuable comments. We truly hope that through the development of Visual Patient Avatar ICU we can make a difference in the quality of patient care. At the moment, Visual Patient is only implemented in University Hospital Zurich and Bonn. This does only include basic option without Visual Patient Avatar ICU. We hope that with further work and developments, Visual Patient Avatar ICU can be implemented across multiple hospitals in daily practice and long term observational studies for mortality and morbidity studies can be conducted.
We hope that you will find our revised paper suitable for publication in your journal and look forward to hearing back from you soon.
With kind regards,
Dr. med. Justyna Lunkiewicz
Reviewer 2 Report
Comments and Suggestions for Authors
The paper with title "User Perceptions of Avatar-Based Patient Monitoring for Intensive Care Units: An International Exploratory Sequential Mixed-Methods Study" is an interesting paper related to user perception of medical staff for using Cisual Patient Avatar ICU.
In the paper appear in the text at line 131 - Appendix 1. At the end of the paper in the section Supplementary Materials the link to Appendix 1 is not working (https://www.mdpi.com/xxx/s1)
The authors should add more details about the online survey in section 2.4.2. Part II: Online Survey, what questions was used ? why the authors used SurveyMonkey and not other tool?
In the section 2.5. Statistical Analysis the authors what they used for their study, but for better understanding they should give some examples.
In section 3.2.2 and 3.2.3 are about positive and negative statement, after this the authors should add a summary in a table format of the result for a better understanding.
Author Response
Dear Reviewer,
Thank you for the valuable and constructive feedback. Hereby I respond to the reviewers’ comments and resubmit the revised manuscript having incorporated required adjustments.
Feedback nr 1
In the paper appear in the text at line 131 - Appendix 1. At the end of the paper in the section Supplementary Materials the link to Appendix 1 is not working (https://www.mdpi.com/xxx/s1)
Author`s reply: Thank you for your insight. File in Appendix 1 will be reuploaded in the MDPI software as well as the file is attached to the Author`s reply comment box.
Feedback nr 2
The authors should add more details about the online survey in section 2.4.2. Part II: Online Survey, what questions was used ? why the authors used SurveyMonkey and not other tool?
Author`s reply: Figure 4 (Results section 3.3. Part II: Online survey) contains five statements used in the online survey. The statements were: Visual Patient`s ICU installation display is helpful.; Visual Patient ICU provides a better overview.; Sometimes Visual Patient ICU is overloaded with information.; Although Visual Patient ICU is intuitive, I would need training before clinical use.; In a critical situation, Visual Patient ICU enabled me to recognize the pathological deviations more quickly.
There statements are now also included to the main text of the manuscript in Method section 2.4.2. Part II Online survey.
As SurveyMonkey is a reliable, user-friendly and internationally known platform for survey conduction, we believed it suited best the international-based methodology of our study.
Feedback nr 3
In the section 2.5. Statistical Analysis the authors what they used for their study, but for better understanding they should give some examples.
Author`s reply: Thank you for drawing our attention to a potential difficulty with understanding of our statistical analysis. We have added two references to the section 2.5 Statistical Analysis:
- McHugh, M.L., Interrater reliability: the kappa statistic. Biochem Med (Zagreb), 2012. 22(3): p. 276-82.
- Landis, J.R. and G.G. Koch, The measurement of observer agreement for categorical data. Biometrics, 1977. 33(1): p. 159-74.
To keep the overview of the manuscript we have intentionally omitted examples in the method section The detailed examples are described in the results and discussion section.
Feedback nr 4
In section 3.2.2 and 3.2.3 are about positive and negative statement, after this the authors should add a summary in a table format of the result for a better understanding.
Author`s reply: Table 2 includes the major topics and subthemes as well as selected examples of the statements. The field notes obtained from 50 participants gave a total of 148 statements. The percentages refer to all codable statements (N= 139). Due to high number of statements, selected examples were included in already large Table 2 in the main body of the manuscript. For all statements, please refer to the Appendix. Appendix includes translated field notes (in English) of 50 participants of this study.
We hope that you will find our revised paper suitable for publication in your journal and look forward to hearing back from you soon.
With kind regards,
Dr. med. Justyna Lunkiewicz